# Imputation of spatially-resolved transcriptomes by graph-regularized tensor completion

**Zhuliu Li[1], Tianci Song[1], Jeongsik Yong[2], Rui Kuang[1]***

**1** Department of Computer Science and Engineering, University of Minnesota Twin Cities, Minneapolis, Minnesota, United States of America, **2** Department of Biochemistry, Molecular Biology, and Biophysics, University of Minnesota Twin Cities, Minneapolis, Minnesota, United States of America

* kuang@umn.edu

**Data Availability Statement:** The source code and data are available at https://github.com/kuanglab/FIST.

**Funding:** ZL, TS, JY and RK were supported by a grant from the National Institutes of Health, USA

## Abstract

High-throughput spatial-transcriptomics RNA sequencing (sptRNA-seq) based on in-situ capturing technologies has recently been developed to spatially resolve transcriptome-wide mRNA expressions mapped to the captured locations in a tissue sample. Due to the low RNA capture efficiency by in-situ capturing and the complication of tissue section preparation, sptRNA-seq data often only provides an incomplete profiling of the gene expressions over the spatial regions of the tissue. In this paper, we introduce a graph-regularized tensor completion model for imputing the missing mRNA expressions in sptRNA-seq data, namely FIST, Fast Imputation of Spatially-resolved transcriptomes by graph-regularized Tensor completion. We first model sptRNA-seq data as a 3-way sparse tensor in genes ($p$-mode) and the ($x$, $y$) spatial coordinates ($x$-mode and $y$-mode) of the observed gene expressions, and then consider the imputation of the unobserved entries or fibers as a tensor completion problem in Canonical Polyadic Decomposition (CPD) form. To improve the imputation of highly sparse sptRNA-seq data, we also introduce a protein-protein interaction network to add prior knowledge of gene functions, and a spatial graph to capture the the spatial relations among the capture spots. The tensor completion model is then regularized by a Cartesian product graph of protein-protein interaction network and the spatial graph to capture the high-order relations in the tensor. In the experiments, FIST was tested on ten 10x Genomics Visium spatial transcriptomic datasets of different tissue sections with cross-validation among the known entries in the imputation. FIST significantly outperformed the state-of-the-art methods for single-cell RNAseq data imputation. We also demonstrate that both the spatial graph and PPI network play an important role in improving the imputation. In a case study, we further analyzed the gene clusters obtained from the imputed gene expressions to show that the imputations by FIST indeed capture the spatial characteristics in the gene expressions and reveal functions that are highly relevant to three different kinds of tissues in mouse kidney.

(R01GM113952). TS and RK were supported by Minnesota Robotics Institute Seed Grants, University of Minnesota. The funders had no role in study design, data collection and analysis, decision to publish, or preparation of the manuscript.

**Competing interests:** The authors have declared that no competing interests exist.

## Author summary

Biological tissues are composed of different types of structurally organized cell units playing distinct functional roles. The exciting new spatial gene expression profiling methods have enabled the analysis of spatially resolved transcriptomes to understand the spatial and functional characteristics of these cells in the context of eco-environment of tissue. Due to the technical limitations, spatial transcriptomics data suffers from only sparsely measured mRNAs by in-situ capture and possibly missing spots in tissue regions that entirely failed fixing and permeabilizing RNAs. Our method, FIST (Fast Imputation of Spatially-resolved transcriptomes by graph-regularized Tensor completion), focuses on the spatial and high-sparsity nature of spatial transcriptomics data by modeling the data as a 3-way gene-by-$(x, y)$-location tensor and a product graph of a spatial graph and a protein-protein interaction network. Our comprehensive evaluation of FIST on ten 10x Genomics Visium spatial genomics datasets and comparison with the methods for single-cell RNA sequencing data imputation demonstrate that FIST is a better method more suitable for spatial gene expression imputation. Overall, we found FIST a useful new method for analyzing spatially resolved gene expressions based on novel modeling of spatial and functional information.

This is a *PLOS Computational Biology* Methods paper.

## Introduction

Dissection of complex genomic architectures of heterogeneous cells and how they are organized spatially in tissue are essential for understanding the molecular and cellular mechanisms underlying important phenotypes. For example, each tumor is a mixture of different types of proliferating cancerous cells with changing genetic materials [1]. The cancer cell sub-populations co-evolve in the micro-environment formed around their spatial locations. It is important to understand the cell-cell interactions and signaling as well as the functioning of each individual cell to develop effective cancer treatment and eradicate all cancer clones at their locations [2]. Conventional gene expression analyses have been limited to low-resolution bulk profiling that measures the average transcription levels in a population of cells. With single-cell RNA sequencing (scRNA-seq) [3–5], single cells are isolated with a capture method such as fluorescence-activated cell sorting (FACS), Fluidigm C1 or microdroplet microfluidics and then the RNAs are captured, reverse transcribed and amplified for sequencing the RNAs barcoded for the individual origin cells [6, 7]. While scRNA-seq is useful for detecting the cell heterogeneity in a tissue sample, it does not provide the spatial information of the isolated cells. To map cell localization, earlier in-situ hybridization methods such as FISH [8], FISSEQ [9], smFISH [10] and MERFISH [11] were developed to profile up to a thousand targeted genes in pre-constructed references with single-molecule RNA imaging. Based on in-situ capturing technologies, more recent spatial transcriptomics RNA sequencing (sptRNA-seq) [12–15] combines positional barcoded arrays and RNA sequencing with single-cell imaging to spatially resolve RNA expressions in each measured spot in the spatial array [12, 16–18]. These new technologies have transformed the transcriptome analysis into a new paradigm for connecting single-cell molecular profiling to tissue micro-environment and the dynamics of a tissue region [19–21].

With in-situ capturing technology, RNAs are captured and sequenced in the spots on the spatial genomic array aligned to the locations on the tissue. For example, spatial transcriptomics

technology based on 10x Genomics Visium kit reports the number of copies of RNAs by counting unique molecular identifiers (UMIs) in the read-pairs mapped to each gene [22]. There are still significant technical difficulties. First, in-situ capturing has a low RNA capture efficiency. The earlier spatial transcriptomics technology's detection efficiency is as low as 6.9% and 10x Genomics Visium has only a slightly improved efficiency [23]. In addition, the sample preparation requires highly specific handling of tissue sections. The spots in some tissue regions might entirely fail to fix and permeabilize RNAs due to various possible issues in preparing tissue sections. A few examples of such regions are shown in S1 Fig. Thus, sptRNA-seq data often only provides an incomplete profiling of the gene expressions over the spatial regions of the tissue. Similarly, in scRNA-seq data analysis, the missing gene expressions are called dropout events, which refer to the false quantification of a gene as unexpressed due to the failure in amplifying the transcripts during reverse-transcription [24]. It has been shown in previous studies on scRNA-seq data that normalizations will not address the dropout effects [22, 25]. In the literature, many imputation methods such as Zero-inflated factor analysis (ZIFA) [26], Zero-Inflated Negative Binomial-based Wanted Variation Extraction (ZINB-WaVE) [27] and BISCUIT [25] have been developed to impute scRNA-seq. While these methods are also applicable to impute the spatial gene expressions, they ignore a unique characteristic of sptRNA-seq data, which is the spatial information among the gene expressions in the spatial array, and do not fully take advantage of the functional relations among genes for more reliable joint imputation.

To provide a more suitable method for imputation of spatially-resolved gene expressions, we introduce FIST, Fast Imputation of Spatially-resolved transcriptomes by graph-regularized Tensor completion. FIST is a tensor completion model regularized by a product graph as illustrated in Fig 1. FIST models sptRNA-seq data as a 3-way sparse tensor in genes ($p$-mode) and the $(x, y)$ spatial coordinates ($x$-mode and $y$-mode) of the observed gene expressions (Fig 1A). As shown in Fig 1B, a protein-protein interaction network models the interactions between pairs of genes in the gene mode, and the spatial graph is modeled by a product graph of two chain graphs for columns ($x$-mode) and rows ($y$-mode) in the grid to capture the spatial

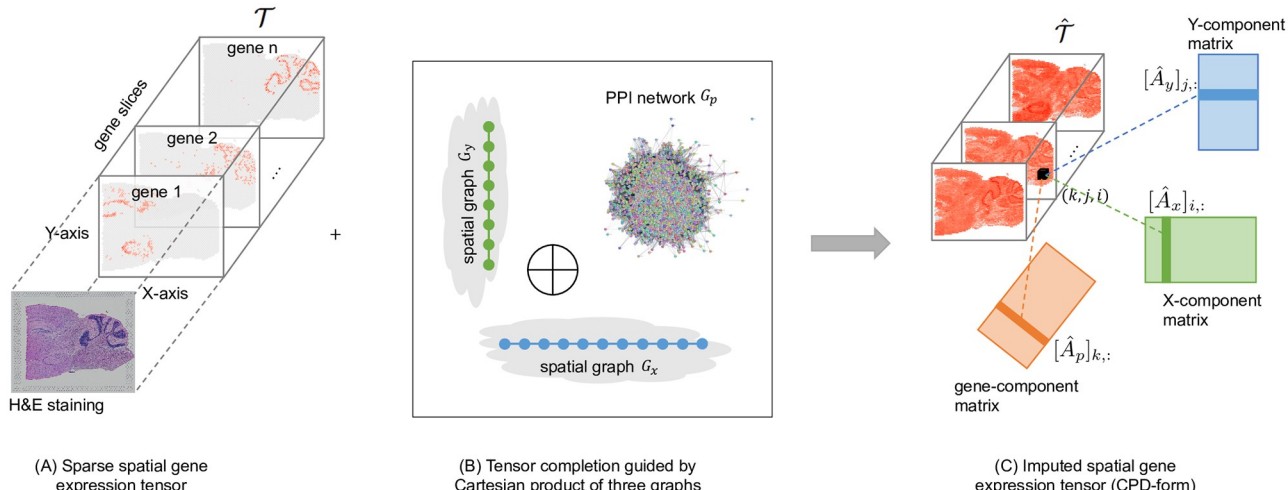

(A) Sparse spatial gene expression tensor

(B) Tensor completion guided by Cartesian product of three graphs

(C) Imputed spatial gene expression tensor (CPD-form)

**Fig 1. Imputation of spatial transcriptomes by graph-regularized tensor completion.** (A) The input sptRNA-seq data is modeled by a 3-way sparse tensor in genes ($p$-mode) and the $(x, y)$ spatial coordinates ($x$-mode and $y$-mode) of the observed gene expressions. H&E image is also shown to visualize the cell morphologies aligned to the spots. (B) A protein-protein interaction network and a spatial graph are integrated as a product graph for tensor completion. The spatial graph is also a product graph of two chain graphs for columns ($x$-mode) and rows ($y$-mode) in the grid. (C) After the imputation, the CPD form of the complete tensor can be used to impute any missing gene expressions, e.g. the entry $(k, j, i)$ can be reconstructed as the sum of the element-wise multiplications of the three components $[\hat{A}_p]_{k,:}$, $[\hat{A}_y]_{j,:}$ and $[\hat{A}_x]_{i,:}$.

relations among the $(x, y)$ spots. The Cartesian product of these graphs with prior knowledge of gene functions and the spatial relations among the capture spots are then introduced as a regularization of tensor completion to obtain the Canonical Polyadic Decomposition (CPD) of the tensor. The imputation of the unobserved entries can then be derived by reconstructing the entries in the completed tensor shown in Fig 1C. In the experiments, we comprehensively evaluated FIST on ten 10x Genomics Visium spatial genomics datasets by comparison with widely used methods for single-cell RNA sequencing data imputation. We also analyzed a mouse kidney dataset with more functional interpretation of the gene clusters obtained by the imputed gene expressions to detect highly relevant functions in the clusters expressed in three kidney tissue regions, cortex, outer stripe of the outer medulla (OSOM) and inner stripe of the outer medulla (ISOM).

## Materials and methods

In this section, we first describe the task of spatial gene expression imputation, and next introduce the mathematical model for graph-regularized tensor completion problem. We then present a fast iterative algorithm FIST to solve the optimization problem defined to optimize the model. We also provide the convergence analysis of proposed algorithm in S1 File. Finally, we provide a review of several state-of-the-art methods for scRNA-seq data imputation, which are also compared in the experiments later. The notations which will be used for the derivations in the forthcoming sections are summarized in Table 1.

### Imputation of spatial gene expressions by tensor modeling

Let $\mathcal{T} \in \mathbb{R}_+^{n_p \times n_y \times n_x}$ be the 3-way sparse tensor of the observed spatial gene expression data as show in Fig 1A, with the missing gene expressions represented as zeros, where $n_p$ denote the total number of genes, $n_x$ and $n_y$ denote the dimensions of the $x$ and $y$ spatial coordinates of the spatial transcriptomics array. Our goal is to learn a complete spatial gene expression tensor $\hat{\mathcal{T}} \in \mathbb{R}_+^{n_p \times n_y \times n_x}$ from $\mathcal{T}$ as illustrated in Fig 1C. However, it is computationally expensive and often infeasible to compute or store a dense tensor $\hat{\mathcal{T}}$, especially in high spatial resolutions with millions of spots. Therefore, we propose to compute an economy-size representation of

**Table 1. Notations.**

| Notation | Definition |
| --- | --- |
| $G_x, G_y$ | Spatial chain graphs of $(x, y)$ coordinates |
| $G_p$ | Protein-protein interaction (PPI) network |
| $n_x, n_y, n_p$ | Number of vertices in $G_x, G_y, G_p$ |
| $W_x \in \mathbb{R}_{[0,1]}^{n_x \times n_x}, W_y \in \mathbb{R}_{[0,1]}^{n_y \times n_y}, W_p \in \mathbb{R}_{[0,1]}^{n_p \times n_p}$ | Adjacency matrix of $G_x, G_y, G_p$ |
| $L_x \in \mathbb{R}^{n_x \times n_x}, L_y \in \mathbb{R}^{n_y \times n_y}, L_p \in \mathbb{R}^{n_p \times n_p}$ | Graph Laplacian of $G_x, G_y, G_p$ |
| $\mathfrak{G}(x, y, p)$ | Cartesian product of $G_x, G_y, G_p$ |
| $\mathfrak{W}(x, y, p) \in \mathbb{R}_{[0,1]}^{n_x n_y n_p \times n_x n_y n_p}$ | Adjacency matrix of $\mathfrak{G}(x, y, p)$ |
| $\mathfrak{L}(x, y, p) \in \mathbb{R}^{n_x n_y n_p \times n_x n_y n_p}$ | Graph Laplacian of $\mathfrak{G}(x, y, p)$ |
| $\mathcal{T} \in \mathbb{R}_+^{n_p \times n_y \times n_x}$ | Incomplete spatial gene expression tensor |
| $\hat{\mathcal{T}} \in \mathbb{R}_+^{n_p \times n_y \times n_x}$ | Complete spatial gene expression tensor |
| $\mathcal{M} \in \mathbb{R}_{[0,1]}^{n_p \times n_y \times n_x}$ | Binary mask tensor |
| $\hat{A}_x \in \mathbb{R}_+^{n_x \times r}, \hat{A}_y \in \mathbb{R}_+^{n_y \times r}, \hat{A}_p \in \mathbb{R}_+^{n_p \times r}$ | CPD component matrices of $\hat{\mathcal{T}}$ |
| $\mathbf{vec}(\mathcal{T}) \in \mathbb{R}^{n_x n_y n_p \times 1}$ | Rearrange $\mathcal{T}$ to be a vector |

$\hat{\mathcal{T}}$ via an equality constraint $\hat{\mathcal{T}} = [\![\hat{A}_p, \hat{A}_y, \hat{A}_x]\!]$, which is called *Canonical Polyadic Decomposition (CPD)* [28] of $\hat{\mathcal{T}}$ defined below

$$\hat{\mathcal{T}} = [\![\hat{A}_p, \hat{A}_y, \hat{A}_x]\!] = \sum_{i=1}^{r} [\hat{A}_p]_{:,i} \circ [\hat{A}_y]_{:,i} \circ [\hat{A}_x]_{:,i}, \qquad (1)$$

where $r$ is the rank of $\hat{\mathcal{T}}$, and $\circ$ denotes the vector outer product. Here, $[\hat{A}_x]_{:,i}$ is the $i$-th column of the low-rank matrix $\hat{A}_x \in \mathbb{R}^{n_x \times r}$, which can be similarly defined for $[\hat{A}_y]_{:,i}$ and $[\hat{A}_p]_{:,i}$. By utilizing the tensor CPD form, we replaced the optimization variables from $\hat{\mathcal{T}}$ to $\hat{A}_p$, $\hat{A}_y$ and $\hat{A}_x$, reducing the number of parameters from $n_p n_y n_x$ to $r(n_p + n_y + n_x)$. The advantage of the tensor representation is to incorporate the 2-D spatial $x$-mode and $y$-mode such that the grid structure is preserved within the columns and the rows of the spatial array in the tensor, which contains useful spatial information. Next, we introduce the tensor completion model over $\hat{A}_p$, $\hat{A}_y$ and $\hat{A}_x$.

## Graph regularized tensor completion model

The key ideas of modeling the task of spatial gene expression imputation are i) the inferred complete spatial gene expression tensor $\hat{\mathcal{T}}$ is regularized to integrate the spatial arrangements of the spots in the tissue array and the functional relations among the genes; ii) the observed part in $\mathcal{T}$ is also required to be preserved in $\hat{\mathcal{T}}$ as the completion task requires; and iii) the inferred tensor $\mathcal{T}$ is compressed as the economy-size representation $\hat{\mathcal{T}} = [\![\hat{A}_p, \hat{A}_y, \hat{A}_x]\!]$ for scalable space and time efficiencies. The novel optimization formulation is shown below in Proposition 1,

**Proposition 1**. *The complete spatial gene expression tensor $\hat{\mathcal{T}} \in \mathbb{R}^{n_p \times n_y \times n_x}$ can be obtained by solving the following optimization problem*:

$$\begin{aligned} \underset{\{\hat{A}_p, \hat{A}_y, \hat{A}_x\}}{minimize} \quad & \frac{1}{2}||\mathcal{M} \circledast (\mathcal{T} - \hat{\mathcal{T}})||_{\mathcal{F}}^2 + \frac{\lambda}{2}\mathbf{vec}(\hat{\mathcal{T}})^T \mathfrak{L}(x,y,p)\mathbf{vec}(\hat{\mathcal{T}}) \\ subject\ to \quad & \hat{\mathcal{T}} = [\![\hat{A}_p, \hat{A}_y, \hat{A}_x]\!] \\ & \hat{A}_p \geq 0, \hat{A}_x \geq 0, \hat{A}_y \geq 0. \end{aligned} \qquad (2)$$

*where $\lambda \in [0, 1]$ is a model hyperparameter; $\circledast$ denotes the Hadamard product; and $||.||_{\mathcal{F}}$ denotes the Frobenius norm of a tensor.*

There are two optimization terms in the model defined in Eq (2), consistency with the observations (the first term) and Cartesian product graph regularization (the second term), which are explained below,

- **Consistency with the observations**
  We introduce a binary mask tensor $\mathcal{M}$ to indicate the indices of the observed entries in $\mathcal{T}$. The $(i, j, k)$-th entry $\mathcal{M}_{i,j,k}$ which is defined below, represents whether the $(i, j, k)$-th element in $\mathcal{T}$ is observed or not.

$$\mathcal{M}_{i,j,k} = \begin{cases} 1 & \text{if } \mathcal{T}_{i,j,k} \text{ is observed,} \\ 0 & \text{otherwise.} \end{cases}$$

By introducing the squared-error in $\mathcal{F}$-norm $||\mathcal{M} \circledast (\mathcal{T} - \hat{\mathcal{T}})||_{\mathcal{F}}^2$ in the model, we ensure

the inferred spatial gene expression tensor $\hat{\mathcal{T}}$ is consistent with its observed counterparts in $\mathcal{T}$.

- **Cartesian product graph regularization**

  Two useful assumptions to introduce prior knowledge for inferring the tensor are *1) the spatially adjacent spots should share similar gene expressions, and 2) the expressions of two genes are likely highly correlated if they share similar gene functions* [29, 30]. We introduce a spatial graph and a protein-protein interaction (PPI) network into the model.

  We first encode the spatial information in two undirected unweighted chain graphs $G_x = (V_x, E_x)$ and $G_y = (V_y, E_y)$, where $V_x$ and $V_y$ are vertex sets and $E_x$ and $E_y$ are edge sets. There are $|V_x| = n_x$ vertices in $G_x$ where $n_x$ is the number of the spatial coordinates along the $x$-axis of the spatial array. Two vertices in $G_x$ are connected by an edge if they are adjacent along the $x$-axis. The connections in $G_y$ can be similarly defined to encode the $y$-coordinates of the tissue. We also incorporate the topological information of a PPI network downloaded from Bio-GRID 3.5 [31] to use the functional modules in the PPI network. We denote the PPI network as $G_p = (V_p, E_p)$ which contains $|E_p|$ experimentally documented physical interactions among the $|V_p| = n_p$ proteins. We then use the Cartesian product graph [32] $\mathfrak{G}(x, y, p) = (V, E)$ of the three individual graphs $G_x$, $G_y$ and $G_p$ to regularize the elements in $\hat{\mathcal{T}}$, where $|V| = n_x n_y n_p$. The $(v_x v_y v_p)$-th vertex in $V$ represents a triple of vertices $\{v_x \in V_x, v_y \in V_y, v_p \in V_p\}$ from each of the three graphs. The $(a_x, a_y, a_p)$-th and $(b_x, b_y, b_p)$-th vertices in $V$ are connected by an edge iff for any $i, j \in \{x, y, p\}$, $(a_i, b_i) \in E_i$ and $a_j = b_j \in V_j$ for all $j \neq i$. For a graph $G_i = (V_i, E_i)$ where $i \in \{x, y, p\}$, we denote its adjacency matrix as $W_i$, degree matrix as $D_i$, and graph Laplacian matrix as $L_i = D_i - W_i$. The adjacency and graph Laplacian matrices of $\mathfrak{G}(x, y, p)$ are obtained as $\mathfrak{W}(x, y, p) = W_x \oplus W_y \oplus W_p$ and $\mathfrak{L}(x, y, p) = L_x \oplus L_y \oplus L_p$ respectively, where $\oplus$ denotes the Kronecker sum [33].

  By introducing the term $\mathbf{vec}(\hat{\mathcal{T}})^T \mathfrak{L}(x, y, p) \mathbf{vec}(\hat{\mathcal{T}})$ in Eq (2), the inferred gene expression values in $\hat{\mathcal{T}}$ are ensured to be smooth over the manifolds of the product graph $\mathfrak{G}(x, y, p)$, such that a pair of tensor entries $\hat{\mathcal{T}}_{a_p, a_y, a_x}$ and $\hat{\mathcal{T}}_{b_p, b_y, b_x}$ share similar values if the $(a_x, a_y, a_p)$-th and $(b_x, b_y, b_p)$-th vertices are connected in $\mathfrak{G}(x, y, p)$. A connection implies that the $x$-coordinate $a_x$ and $b_x$ is adjacent or $y$-coordinate $a_y$ and $b_y$ is adjacent or gene $a_p$ and gene $b_p$ are connected in the PPI, with the two other dimensions fixed. Using Cartesian product graph is a more conservative strategy to connect multi-relations in a high-order graph as we have shown in [34] since only replacing one of the dimensions by the immediate neighbors is allowed to create connections. Note that it also possible to use tensor product graph or strong product graph [34] but there could be too many connections to provide meaningful connectivity in the product graph for helpful regularization. It is known that genes' connectivities in PPI network correlate with their co-expressions. We also justified this hypothesis on the spatial transcriptomics data by examining the relation between the connectivity in PPI network and the co-expression in spatial locations among the genes in the 10 different 10x Genomics Visium spatial genomics datasets used in this study. The results of this analysis are shown in S2 Fig. We observed higher co-expressions between the genes that are connected with less hops in the PPI, which clearly supports our assumptions.

## FIST Algorithm

The model introduced in Eq (2) is non-convex on variables $\{\hat{A}_p, \hat{A}_y, \hat{A}_x\}$ jointly, thus finding its global minimum is difficult. In this section, we propose an efficient iterative algorithm **F**ast **I**mputation of **S**patially-resolved transcriptomes by graph-regularized **T**ensor completion

(FIST) to find its local optimal solution using the multiplicative updating rule [35], based on derivatives of $\hat{A}_p$, $\hat{A}_y$ and $\hat{A}_x$. Without loss of generality, we only show the derivations with respect to $\hat{A}_p$, and provide the FIST algorithm in Algorithm 1.

We first bring the equality constraint $\hat{\mathcal{T}} = [\![\hat{A}_p, \hat{A}_y, \hat{A}_x]\!]$ in Eq (2) into the objective function, and rewrite the objective function as

$$
\begin{aligned}
\mathcal{J} &= \mathcal{J}_1 + \lambda \mathcal{J}_2 \\
\mathcal{J}_1 &= \frac{1}{2} ||\mathcal{M} \circledast (\mathcal{T} - [\![\hat{A}_p, \hat{A}_y, \hat{A}_x]\!])||_{\mathcal{F}}^2 \\
\mathcal{J}_2 &= \frac{1}{2} \mathbf{vec}([\![\hat{A}_p, \hat{A}_y, \hat{A}_x]\!])^T \mathfrak{L}(x,y,p) \mathbf{vec}([\![\hat{A}_p, \hat{A}_y, \hat{A}_x]\!])
\end{aligned}
\tag{3}
$$

The partial derivative of $\mathcal{J}_1$ with respect to $\hat{A}_p$ can be computed as

$$
\frac{\partial \mathcal{J}_1}{\partial \hat{A}_p} = (\mathcal{M}_{(1)} \circledast \hat{\mathcal{T}}_{(1)} - \mathcal{M}_{(1)} \circledast \mathcal{T}_{(1)})(\hat{A}_x \odot \hat{A}_y),
\tag{4}
$$

where $\mathcal{T}_{(1)} \in \mathbb{R}^{n_p \times n_x n_y}$ denotes the matrix flattened from tensor $\mathcal{T}$; $\odot$ denotes the Khatri–Rao product [28]. Note that the term $\mathcal{M}_{(1)} \circledast \hat{\mathcal{T}}_{(1)}$ in Eq (4) implies we only need to compute the entries in $\hat{\mathcal{T}}$ which correspond to the non-zero entries (indices of the observed gene expression) in $\mathcal{M}$. The rest of the computation in Eq (4) involves the well-known MTTKRP (matricized tensor times Khatri-Rao product) [36] operation, which is in the form of $\mathcal{X}_{(1)}(\hat{A}_x \odot \hat{A}_y)$, and can be computed in $O(r|\mathcal{X}|)$ if $\mathcal{X}$ is a sparse tensor with $|\mathcal{X}|$ non-zeros, and $\hat{A}_x$ and $\hat{A}_y$ have $r$ columns. Thus, the overall time complexity of computing Eq (4) is $O(r|\mathcal{M}|)$.

Following the derivations in [34], we obtain the partial derivatives of the second term $\mathcal{J}_2$ as

$$
\frac{\partial \mathcal{J}_2}{\partial \hat{A}_p} = \hat{A}_p(\Phi_x \circledast \Theta_y + \Phi_y \circledast \Theta_x) + L_p \hat{A}_p(\Phi_x \circledast \Phi_y),
\tag{5}
$$

where $\Phi_i = \hat{A}_i^T \hat{A}_i$, and $\Theta_i = \hat{A}_i^T L_i \hat{A}_i$, for all $i \in \{x, y, p\}$. It is not hard to show that the complexity of computing Eq (5) is $O(\sum_{i \in \{x,y,p\}}(r^2 n_i + r n_i^2))$.

Next, we combine $\frac{\partial \mathcal{J}_1}{\partial \hat{A}_p}$ and $\frac{\partial \mathcal{J}_2}{\partial \hat{A}_p}$ to obtain the overall derivative as

$$
\begin{aligned}
\frac{\partial \mathcal{J}}{\partial \hat{A}_p} &= \frac{\partial \mathcal{J}_1}{\partial \hat{A}_p} + \lambda \left( \frac{\partial \mathcal{J}_2}{\partial \hat{A}_p} \right) \\
&= \left[ \frac{\partial \mathcal{J}_1}{\partial \hat{A}_p} \right]^+ - \left[ \frac{\partial \mathcal{J}_1}{\partial \hat{A}_p} \right]^- + \lambda \left( \left[ \frac{\partial \mathcal{J}_2}{\partial \hat{A}_p} \right]^+ - \left[ \frac{\partial \mathcal{J}_2}{\partial \hat{A}_p} \right]^- \right),
\end{aligned}
\tag{6}
$$

where $\left[ \frac{\mathcal{J}_i}{\partial A_p} \right]^+$ and $\left[ \frac{\mathcal{J}_i}{\partial A_p} \right]^-$ are non-negative components in $\frac{\mathcal{J}_i}{\partial A_p}$, which are defined below,

$$
\left[ \frac{\partial \mathcal{J}_1}{\partial \hat{A}_p} \right]^+ = (\mathcal{M}_{(1)} \circledast \hat{\mathcal{T}}_{(1)})(\hat{A}_x \odot \hat{A}_y),
\tag{7}
$$

$$
\left[ \frac{\partial \mathcal{J}_1}{\partial \hat{A}_p} \right]^- = (\mathcal{M}_{(1)} \circledast \mathcal{T}_{(1)})(\hat{A}_x \odot \hat{A}_y),
\tag{8}
$$

$$\left[\frac{\partial \mathcal{J}_2}{\partial \hat{A}_p}\right]^{+} = \hat{A}_p(\Phi_x \circledast \Theta_y^D + \Phi_y \circledast \Theta_x^D) + D_p\hat{A}_p(\Phi_x \circledast \Phi_y), \tag{9}$$

$$\left[\frac{\partial \mathcal{J}_2}{\partial \hat{A}_p}\right]^{-} = \hat{A}_p(\Phi_x \circledast \Theta_y^W + \Phi_y \circledast \Theta_x^W) + W_p\hat{A}_p(\Phi_x \circledast \Phi_y), \tag{10}$$

where $\Theta_i^D = \hat{A}_i^T D_i \hat{A}_i$ and $\Theta_i^W = \hat{A}_i^T W_i \hat{A}_i$, for all $i \in \{x, y, p\}$. According to Eq (6), the objective function $\mathcal{J}$ objective will monotonically decrease under the following multiplicative updating rule,

$$[\hat{A}_p]_{a,b} \leftarrow [\hat{A}_p]_{a,b} \left( \frac{\left[\frac{\partial \mathcal{J}_1}{\partial \hat{A}_p}\right]^{-}_{a,b} + \lambda \left[\frac{\partial \mathcal{J}_2}{\partial \hat{A}_p}\right]^{-}_{a,b}}{\left[\frac{\partial \mathcal{J}_1}{\partial \hat{A}_p}\right]^{+}_{a,b} + \lambda \left[\frac{\partial \mathcal{J}_2}{\partial \hat{A}_p}\right]^{+}_{a,b}} \right), \tag{11}$$

where $[\hat{A}_p]_{a,b}$ denotes the $(a, b)$-th element in matrix $\hat{A}_p$. Similarly, we can derive the update rule for $[\hat{A}_x]_{a,b}$ and $[\hat{A}_p]_{a,b}$ as follows,

$$[\hat{A}_y]_{a,b} \leftarrow [\hat{A}_y]_{a,b} \left( \frac{\left[\frac{\partial \mathcal{J}_1}{\partial \hat{A}_y}\right]^{-}_{a,b} + \lambda \left[\frac{\partial \mathcal{J}_2}{\partial \hat{A}_y}\right]^{-}_{a,b}}{\left[\frac{\partial \mathcal{J}_1}{\partial \hat{A}_y}\right]^{+}_{a,b} + \lambda \left[\frac{\partial \mathcal{J}_2}{\partial \hat{A}_y}\right]^{+}_{a,b}} \right), \tag{12}$$

$$[\hat{A}_x]_{a,b} \leftarrow [\hat{A}_x]_{a,b} \left( \frac{\left[\frac{\partial \mathcal{J}_1}{\partial \hat{A}_x}\right]^{-}_{a,b} + \lambda \left[\frac{\partial \mathcal{J}_2}{\partial \hat{A}_x}\right]^{-}_{a,b}}{\left[\frac{\partial \mathcal{J}_1}{\partial \hat{A}_x}\right]^{+}_{a,b} + \lambda \left[\frac{\partial \mathcal{J}_2}{\partial \hat{A}_x}\right]^{+}_{a,b}} \right). \tag{13}$$

We then propose an efficient iterative algorithm FIST in Algorithm 1 to find the local optimum of the proposed graph regularized tensor completion problem with time complexity $O(r|\mathcal{M}| + \sum_{i \in \{x,y,p\}}(r^2 n_i + r n_i^2))$. FIST takes the incomplete spatial gene expression tensor $\mathcal{T}$, PPI network and spatial chain graphs as input (line 1-2 in Algorithm 1), and outputs the inferred CPD representation of the complete spatial gene expression tensor $\hat{\mathcal{T}}$ (line 9 in Algorithm 1), via solving the optimization problem defined in Proposition 1 with the multiplicative updating rule (line 5-7 in Algorithm 1) based on the tensor calculus in Eqs (7)–(10). With the efficient tensor computation in Eqs (7)–(10), the algorithm can avoid computing the full Cartesian product graph and tensors, and break down the calculus into the computation on the individual graphs and the sparse tensors. Therefore, FIST is proven to be a scalable method, which only requires the space $O(|\mathcal{T}| + |\mathcal{M}|)$ to store the sparse tensors, $O(\sum_{i \in \{x,y,p\}} |E_i|)$ to store the graphs, and $O(\sum_{i \in \{x,y,p\}} r n_i)$ to store the factor matrices. Thus, the overall space complexity is $O(|\mathcal{T}| + |\mathcal{M}| + \sum_{i \in \{x,y,p\}}(|E_i| + r n_i))$. The theoretical convergence analysis of FIST is also given in S1 File.

**Algorithm 1** FIST: **F**ast **I**mputation of **S**patially-resolved transcriptomes by graph-regularized **T**ensor completion

```
1: Input: 1) spatial gene expression tensor T ∈ ℝ₊^{n_p×n_y×n_x}, 2) binary mask
tensor M ∈ ℝ_{[0,1]}^{n_p×n_y×n_x} which indicates the observed part in T, 3) protein-
protein interaction (PPI) network G_p and 4) hyper parameter λ.
2: Construct the spatial chain graphs G_x and G_y as described in the
text.
```

```
3: Randomly initialize Âₚ ∈ ℝ₊^{nₚ×r},  Â_y ∈ ℝ₊^{n_y×r} and Â_x ∈ ℝ₊^{n_x×r} as non-negative
matrices.
4: while not converge do
5:    update Âₚ by Eq (11).
6:    update Â_y by Eq (12).
7:    update Â_x by Eq (13).
8: end while
9: Output: the low-rank matrices Âₚ,Â_y and Â_x, which form the CPD
representation of the inferred spatial gene expression tensor
𝒯̂ = ⟦Âₚ,Â_y,Â_x⟧ ∈ ℝ₊^{nₚ×n_y×n_x}.
```

## Related methods for comparison

To benchmark the performance of FIST, we compared it with three matrix factorization (MF)-based methods (with graph regularizations) and a nearest neighbors (NN)-based method, which have been applied to impute various types of biological data including the imputation of dropouts in single-cell RNA sequencing (scRNA-seq) data. Note that NMF-based methods have been shown to be effective for learning latent features and clustering high-dimension sparse genomic data [37].

- **ZIFA**: Zero-inflated factor analysis (ZIFA) [26] factorizes the single cell expression data $Y \in \mathbb{R}^{N \times D}$ where $N$ and $D$ denote the number of single cells and genes respectively, into a factor loading matrix $A \in \mathbb{R}^{K \times D}$ and a matrix $Z \in \mathbb{R}^{N \times K}$ which spans the latent low-dimensional space where dropouts can happen with a probability specified by an exponential decay associated with the expression levels. The imputed matrix can be computed as $\hat{Y} = ZA + \mu$, where $\mu \in \mathbb{R}^{1 \times D}$ is the latent mean vector.

- **REMAP**: Since ZIFA is a probabilistic MF model which does not utilize the spatial and gene networks, we therefore also compare with REMAP [38], which was developed to impute the missing chemical-protein associations for the identification of the genome-wide off-targets of chemical compounds. REMAP factorizes the incomplete chemical-protein interactions matrix into the chemical and protein low-rank matrices, which are regularized by the compound similarity graph and protein sequence similarity (NCBI BLAST [39]) graph respectively.

- **GWNMF**: Both ZIFA and REMAP are only applicable to the spot-by-gene matrix which is a flatten of an input tensor $\mathcal{T}$. Such flattening process assumes the spots are independent from each other, and thus loses the spatial information. To keep the spatial arrangements, we also apply MF to each $n_x \times n_y$ slice in tensor $\mathcal{T}$. Specially, we adopt the graph regularized weighted NMF (GWNMF) [40] method to impute each $n_x \times n_y$ gene slice. We let GWNMF use the same $x$-axis and $y$-axis graphs $G_x$ and $G_y$ as described in the previous section to regularize the MF.

- **Spatial-NN**: It has been observed that in sparse high-dimensional scRNA-seq data, constructing a nearest neighbor (NN) graph among cells can produce more robust clusters in the presence of dropouts because of taking into account the surrounding neighbor cells [41]. Such rationale has be considered in the clustering methods such as Seurat [42] and shared nearest neighbors (SNN)-Cliq [41], and can also be adopted to impute the spatial gene expression data. We introduce a SNN-based baseline Spatial-NN using neighbor averaging to compare with FIST. Specifically, to impute the missing expression of a target spot, Spatial-

NN first searches its spatially nearest spots with observed gene expressions, then assign their average gene expression to the target spot.

We used the provided Python package (https://github.com/epierson9/ZIFA) to experiment with ZIFA, and the provided MATLAB package (https://github.com/hansaimlim/REMAP) to experiment with REMAP. To apply both methods, we rearranged the data tensor $\mathcal{T} \in \mathbb{R}^{n_p \times n_y \times n_x}$ to a matrix $T \in \mathbb{R}^{N \times n_p}$, where $N = n_x n_y$ denotes the total number of spots. The spatial graph of REMAP is constructed by connecting two spots if they are spatially adjacent. REMAP adopts the same PPI network as the gene graph $G_p$ as used by FIST. We used MATLAB to implement GWNMF and Spatial-NN ourselves to impute each gene slice $T_i \in \mathbb{R}^{n_x \times n_y}$ in $\mathcal{T}$. In the comparisons, the graph hyperparameter $\lambda$ of FIST is selected from {0, 0.01, 0.1, 1}. The graph hyperparameters of REMAP and GWNMF are set by searching the grids from {0.1, 0.5, 0.9, 1} and {0, 0.1, 1, 10, 100} respectively as suggested in the original studies. Note that different methods use different scales of graph hyperparameters since the gradients of their variables with respective to the regularization terms are in different scales. The optimal hyper-parameters are selected by the validation set for each method. For FIST, REMAP and GWNMF, we applied PCA on matrix $T \in \mathbb{R}^{N \times n_p}$ to determine the rank $r \in [200, 300]$ of the low-rank factor matrices, such that at least 60% of the variance is accounted for by the top-$r$ PCA components of $T$. The latent dimension $K$ of ZIFA is set to 10 since it is time consuming to run ZIFA with a larger $K$. We also observed that increasing $K$ from 10 to 50 does not show clear improvement on the imputation accuracy.

## Results

In this section, we first describe data preparation and performance measure, and then show the results of spatial gene expression imputation. We also analyzed the results by the gene-wise density of the gene expressions and regularization by permuted graphs. Finally, we analyzed the imputed spatial gene expressions in the Mouse Kidney Section dataset to show several interesting gene clusters revealing functional characteristics of the three tissue regions, cortex, OSOM and ISOM.

### Preparation of spatial gene expression datasets

We downloaded the spatial transcriptomic datasets from 10x Genomics (https://support.10xgenomics.com/spatial-gene-expression/datasets/), which is a collection of spatial gene expressions in 10 different tissue sections from mouse brain, mouse kidney, human breast cancer, human heart and human lymph node as listed in Table 2. All the sptRNA-seq datasets were

**Table 2. 10x Genomics spatial transcriptome data from 10 tissue sections.**

| Dataset | Tissue section | Tensor dimensions | Density |
|---------|----------------|-------------------|---------|
| HBA1 | Human Breast Cancer (Block A Section 1) | $13,426 \times 60 \times 77$ | 0.093 |
| HBA2 | Human Breast Cancer (Block A Section 2) | $13,470 \times 58 \times 75$ | 0.100 |
| HH | Human Heart | $7,487 \times 63 \times 70$ | 0.049 |
| HLN | Human Lymph Node | $12,368 \times 61 \times 78$ | 0.088 |
| MKC | Mouse Kidney Section (Coronal) | $12,264 \times 41 \times 56$ | 0.103 |
| MBC | Mouse Brain Section (Coronal) | $13,570 \times 49 \times 74$ | 0.110 |
| MB1P | Mouse Brain Serial Section 1 (Sagittal-Posterior) | $15,404 \times 62 \times 67$ | 0.115 |
| MB2P | Mouse Brain Serial Section 2 (Sagittal-Posterior) | $12,497 \times 63 \times 65$ | 0.077 |
| MB1A | Mouse Brain Serial Section 1 (Sagittal-Anterior) | $12,658 \times 59 \times 66$ | 0.105 |
| MB2A | Mouse Brain Serial Section 2 (Sagittal-Anterior) | $12,295 \times 63 \times 66$ | 0.082 |

collected with 10x Genomics Visium Spatial protocol (v1 chemistry) [14] to profiles each tissue section with a high density hexagonal array with 4,992 spots to achieve a resolution of 55 μm (1-10 cells per spot). To fit a tensor model on the spatial gene expression datasets, we organized each of the 10 datasets into a 3-way tensor $\mathcal{T} \in \mathbb{R}^{n_p \times n_y \times n_x}$, where the $(i, j, k)$-th entry in $\mathcal{T}$ is the UMI count of the $i$-th gene at the $(k, j)$-th coordinate in the array. Note that the spots are arranged in a perfect grid in earlier spatial transcriptomic arrays and the rows and columns in the grid can be used directly as the coordinates $(n_x, n_y)$. In the Visium array slide, the spots are arranged in a hivegrid. To map the spatial coordinates $(n_x, n_y)$, we shifted the odd-numbered rows by a half of a spot for a convenient arrangement of the spots in the tensor without loss of generality. We set the entries in $\mathcal{T}$ to zeros if their UMI counts is lower than 3. We then removed the genes with no UMI counts across the spots, and removed the empty spots in the boundaries of the four sides in the H&E staining from $\mathcal{T}$. The log-transformation is finally applied to every entry of $\mathcal{T}$ as $\mathcal{T}_{i,j,k} \leftarrow \log(1 + \mathcal{T}_{i,j,k})$. The sizes and densities of the 10 different spatial gene expression tensors after prepossessing are summarized in Table 2. Finally, we downloaded the full Homo sapiens and Mus musculus protein-protein interactions (PPI) networks from BioGRID 3.5 [31] as the gene network $G_p$ to match with the genes in each dataset.

## Evaluations and performance measures

We applied 5-fold cross-validation to evaluate the performance of imputing spatial gene expressions by spatial spots or genes as follows:

- **Spot-wise evaluation**: We chose 4-fold of the non-empty spatial spots for training and validation, and held out the rest 1-fold non-empty spatial spots as test data. When evaluating the expressions $\mathcal{T}_{:,j,k} \in \mathbb{R}^{n_p \times 1}$ in the $(k, j)$-th spatial spot, we set the vectors $\mathcal{T}_{:,j,k}$ and $\mathcal{M}_{:,j,k}$ in the input tensor $\mathcal{T}$ and mask tensor $\mathcal{M}$ to zeros to indicate that the expressions in this spot are unobserved; next, we use the learned low-rank matrices $\hat{A}_p, \hat{A}_y$ and $\hat{A}_x$ to construct the predicted gene expressions $\hat{\mathcal{T}}_{:,j,k}$ as described in Eq (1).

- **Gene-wise evaluation**: For each gene, we chose 4-fold of its observed expressions (non-zeros in $\mathcal{T}$) for training and validation, and held out the rest 1-fold of observed expressions as test data. When evaluating the 1-fold expressions in the $i$-th gene $\mathcal{T}_{i,:,:} \in \mathbb{R}^{n_y \times n_x}$, we set the corresponding entries in $\mathcal{T}_{i,:,:}$ and $\mathcal{M}_{i,:,:}$ in the input tensor $\mathcal{T}$ and mask tensor $\mathcal{M}$ to zeros to indicate the expressions in this fold are unobserved; next, we use the learned low-rank matrices $\hat{A}_p, \hat{A}_y$ and $\hat{A}_x$ to construct the predicted gene expressions $\hat{\mathcal{T}}_{i,:,:}$ as described in Eq (1).

The hyper-parameter λ is optimized by the validation set for FIST and the baseline methods. Denoting vectors $\mathbf{t} \in \mathbb{R}^{n \times 1}$ and $\hat{\mathbf{t}} \in \mathbb{R}^{n \times 1}$ as the true and predicted expressions in the held-out spatial spot $\mathcal{T}_{:,j,k}$ or the held-out entries in gene $\mathcal{T}_{i,:,:}$, the imputation performance is evaluated by the following three metrics,

- MAE (mean absolute error) $= \frac{1}{n}\left(\sum_{i=1}^{n} |\mathbf{t}_i - \hat{\mathbf{t}}_i|\right)$,

- MAPE (mean absolute percentage error) $= \frac{1}{n}\left(\sum_{i=1}^{n} \left|\frac{\mathbf{t}_i - \hat{\mathbf{t}}_i}{\mathbf{t}_i}\right|\right)$,

- $R^2$ (coefficient of determination) $= 1 - \left(\sum_{i=1}^{n} (\mathbf{t}_i - \hat{\mathbf{t}}_i)^2\right)\left(\sum_{i=1}^{n} \left(\mathbf{t}_i - \frac{\sum_{j=1}^{n} \mathbf{t}_j}{n}\right)^2\right)^{-1}$.

We expect a method to achieve smaller MAE and MAPE and larger $R^2$ for better performance.

## FIST significantly improves the accuracy of imputing spatial gene expressions

The performances of FIST and the baseline methods except for ZIFA in the spot-wise evaluation are compared in Fig 2. ZIFA was excluded from this spot-wise evaluation as it does not allow empty rows (spots) in the implementation of its package, and thus is not applicable to the prediction of the held-out test spots. The average performances of all the spatial spots using each of the 10 sptRNA-seq datasets are shown as bar plots. FIST consistently outperforms all the baselines with lower MAE and MAPE, and larger $R^2$ in all the 10 datasets. We further applied right-tailed paired-sample $t$-tests on $R^2$ values to test against the alternative

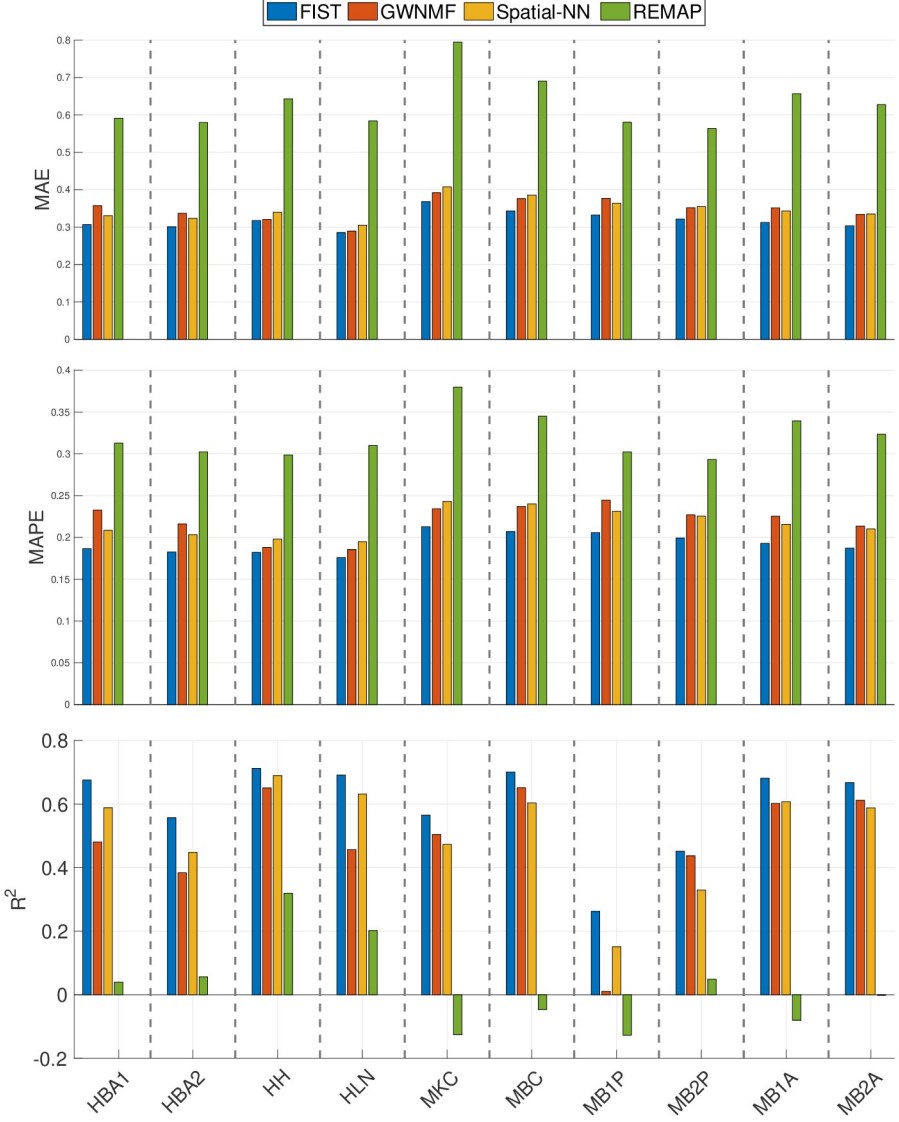

**Fig 2. Spot-wise cross-validation on 10x Genomics data.** The performances of the four compared methods on the 10 tissue sections are measured by 5-fold cross-validation. Each bar shows the mean of the imputation performance of one method on all the spatial spots. The result on each of the 10 datasets is shown in one vertical column separated by dashed lines. The means are also compared between FIST and each of the baseline methods in S1 Table by paired-sample $t$-tests.

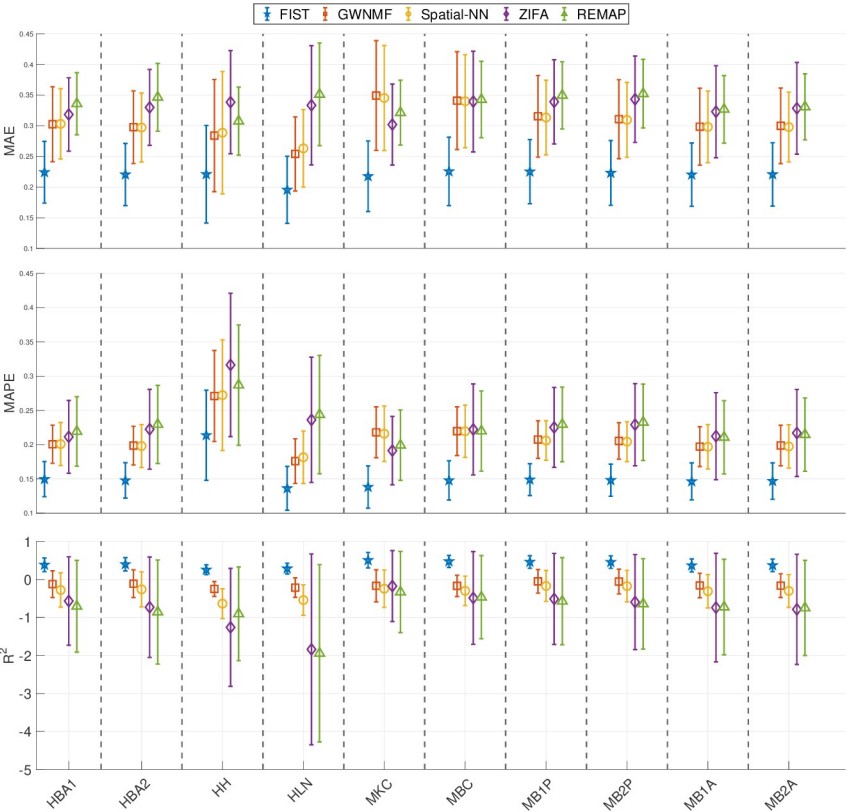

**Fig 3. Gene-wise cross-validation on 10x Genomics data.** The performances of the five compared methods on the 10 tissue sections are measured by 5-fold cross-validation. Each error bar shows the mean and variance of the imputation performance for one method on all the genes. The result on each of the 10 datasets is shown in one vertical column separated by dashed lines.

hypothesis that the $R^2$ produced by FIST has a larger mean than the mean of those produced by each of the baseline methods; and we also applied left-tailed paired-sample $t$-tests on MAE and MAPE values to test against the alternative hypothesis that the MAE and MAPE produced by FIST have a smaller mean than the mean of those produced by each of the baseline methods. The $p$-values in S1 Table show that compared with the baseline methods, FIST has significantly lower MAE and MAPE in each of the 10 datasets, and larger $R^2$ in all comparisons but one, in which FIST performed only slightly better than GWNMF on HB2P dataset by $R^2$.

The performances of FIST and the baseline methods in the gene-wise evaluation are compared in Fig 3. The average and standard deviation of the prediction performances across all the genes are shown as error bar plots in Fig 3. Similar to the spot-wise evaluation, FIST clearly outperforms all the baselines with more robust performances across all the genes, as the variances in all the three evaluation metrics are also lower than the other compared methods. To examine the prediction performance more closely, we also showed the distributions of MAE, MAPE and $R^2$ of individual genes in the 10 datasets in S3–S5 Figs, respectively. The result is consistent with the overall performance in Fig 3. The observations suggest that FIST indeed performs better than the other methods in the imputation accuracy informed by the spatial information in the tensor model. It is also noteworthy that GWNMF, the MF method regularized by the spatial graph applied to each individual gene slice in tensor $\mathcal{T}$, outperforms the other baselines in almost all the datasets. This observation further confirms that the spatial patterns maintained in each gene slice is informative for the imputation task. It is clear that FIST

outperformed GWNMF with better use of the spatial information coupled with the functional modules of the PPI network $G_p$ and the joint imputation of all the genes in the tensor $\mathcal{T}$.

## Cartesian product graph regularization plays a significant role

To demonstrate that the Cartesian product graph regularization in FIST significantly improves the imputation accuracy, we show in Fig 4 the performance of FIST in each of the 10 datasets by varying the graph hyper-parameter λ in the spot-wise evaluation. By increasing λ from 0 to 0.1 to put more belief on the graph information, we observe an appreciable reduction on the MAE and MAPE, and increase on $R^2$ across all the 10 datasets. The observation strongly suggests that the predictions by FIST are improved by leveraging the information carried in the CPG topology, and the belief on the graph information can be effectively optimized by using a validation set in the cross-validation strategy.

To further understand the associations between the CPG regularization and characteristics of the expressions of the genes, we analyzed the genes that are benefiting most from the regularization by the CPG in the gene-wise evaluation. In particular, in the grid search of the optimal λ weight on the CPG regularization term by the validation set, we count the percentage of the genes with optimal λ = 0.01 rather than 0, which means completely ignoring the regularization. To correlate the improved imputations with the sparsity of the gene expressions, we divided all the genes into 4 equally partitioned groups (L1-4) ordered by their densities in the sptRNA-seq data, where L1 and L4 contain the sparsest and the densest gene slices, respectively. For each of the four density levels, we count the percentage of gene slices that benefit from the CPG regularization and plot the results in Fig 5A. In the plots, there is a clear trend that the sparser a gene slice, the more likely it benefits from the CPG regularization in all the 10 datasets. In the densest L4 group, as low as 20% of the genes can benefit from the CPG regularization versus more than 50% in the sparsest L1 group. This is understandable that there is less training information available for sparsely expressed genes (with more dropouts) and the spatial and functional information in the CPG can play a more important role in the imputation by seeking information from the gene's spatial neighbors or the functional neighbors in the PPI network. This observation is also consistent with the fact that the performance of

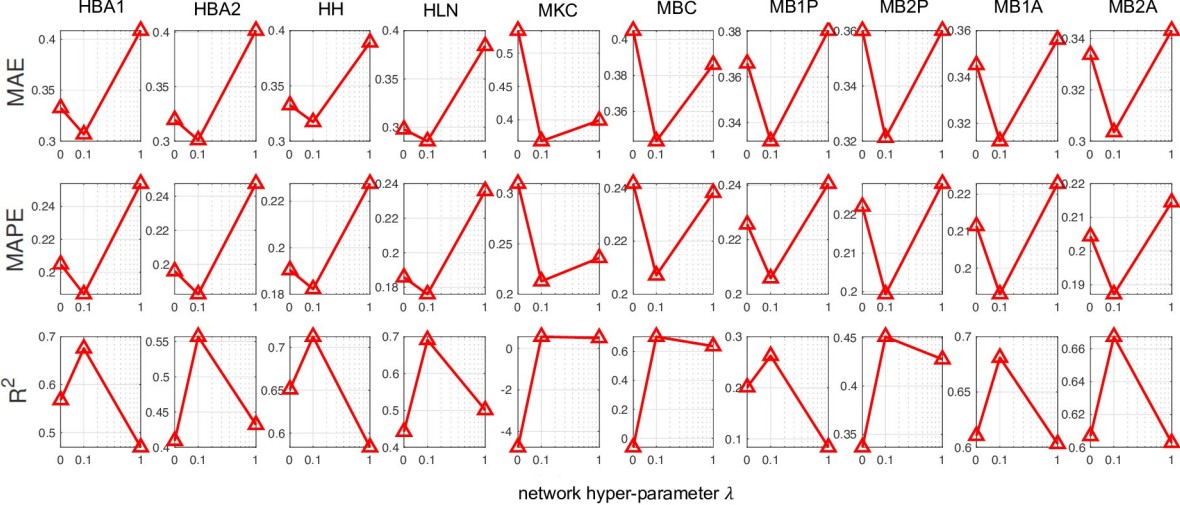

**Fig 4. Analysis of Cartesian product graph regularization with varying network hyper-parameter in spot-wise evaluation.** The plots show the imputation performance of FIST on the ten 10x Genomics datasets with varying network hyper-parameters in {0, 0.1, 1} by MAE, MAPE and $R^2$.

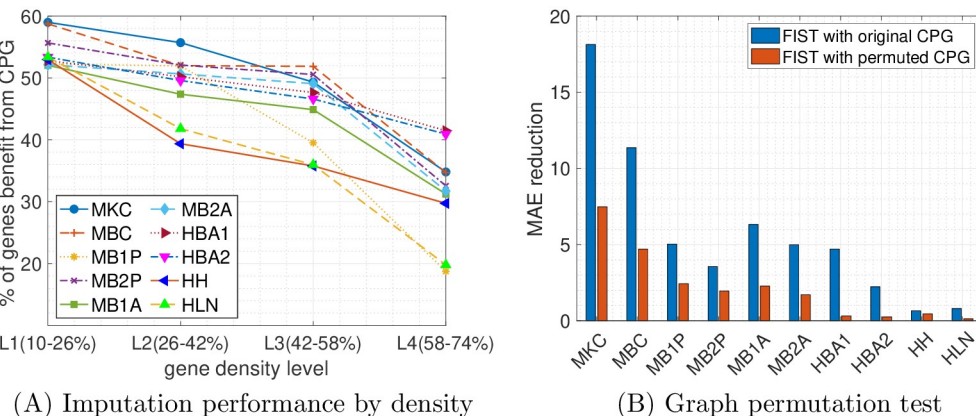

**Fig 5. Analysis of Cartesian product graph regularization on gene-wise evaluation.** (A) The percentages of genes benefit from the CPG are plotted by their densities in four different ranges. Each colored line represents one of the 10 datasets. (B) The total reduction of MAE using the original and permuted graphs are compared across the 10 tissue sections.

tensor completion tends to degrade severely when only a very small fraction of entries are observed [43, 44], and therefore those sparser gene slices tend to benefit more from the side information carried in the CPG.

We also compared the performance of FIST using the CPG of $G_x$, $G_y$ and $G_p$ with the one using a randomly permuted graph from the CPG. To generate the random CPG, we first generated three random graphs by permute $G_x$, $G_y$ and $G_p$ individually which also preserves the degree distributions of the original graphs, by randomly swapping the edges in each graph while keeping the degree of each node. Then we measured the performances of FIST using the original CPG and the CPG obtained from the permuted graphs by MAE reduction, which is the total reduction of MAE on all the genes by varying hyperparameter λ from 0 to 0.01 meaning not using the graph versus using the graph. The comparisons across all the 10 datasets are shown in Fig 5B. We observe that the FIST using the original graphs receives much higher MAE reduction than the FIST using the permuted graphs. This observation suggests that the topology in the original CPG carries rich information that is helpful for the imputation task beyond just the degree distributions preserved in the random graphs.

## FIST imputations recover spatial patterns enriched by highly relevant functional terms

To demonstrate that imputations by FIST can reveal spatial gene expression patterns with highly relevant functional characteristics among the genes in the spatial region, we performed comparative GO enrichment analysis of gene clusters detected with the imputed gene expressions. We conducted a case study on the Mouse Kidney Section data to further analyze the associations between the spatial gene clusters and the relevance between their functional characteristics and three kidney tissue regions, cortex, outer stripe of the outer medulla (OSOM) and inner stripe of the outer medulla (ISOM).

To validate the hypothesis that the imputed sptRNA-seq tensor $\tilde{\mathcal{T}}$ given below

$$\tilde{\mathcal{T}} = (1 - \mathcal{M}) \circledast \hat{\mathcal{T}} + \mathcal{T}$$

can better capture gene functional modules than the sparse sptRNA-seq tensor $\mathcal{T}$ does, we first rearranged both sptRNA-seq tensors into matrices $\tilde{T} \in \mathbb{R}^{N \times n_p}$ and $T \in \mathbb{R}^{N \times n_p}$, where $N = n_x\, n_y$

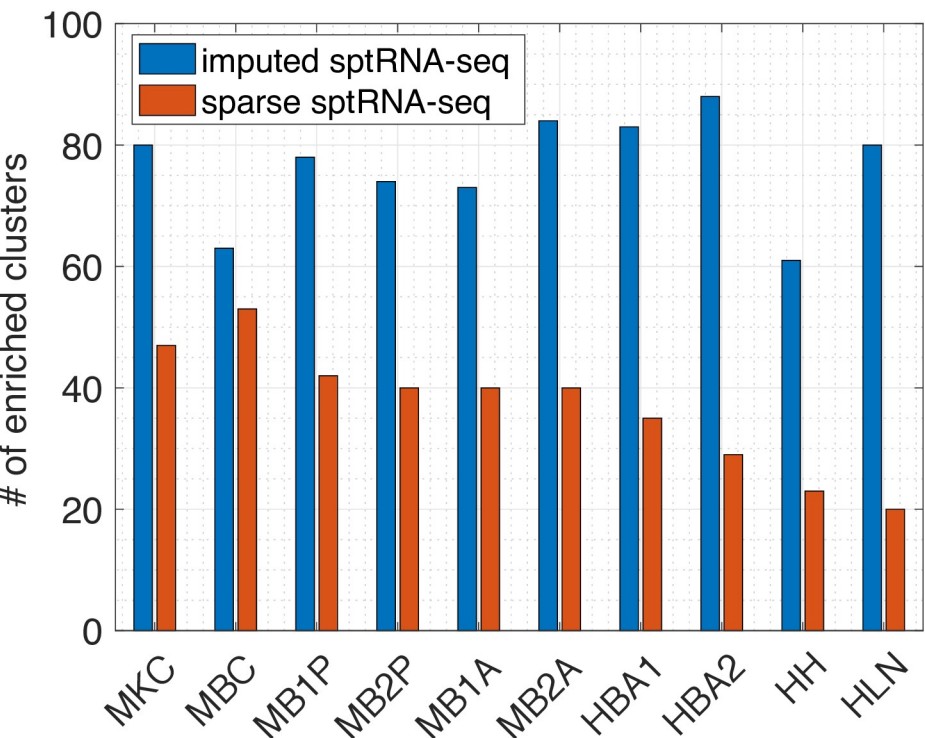

**Fig 6. Enrichment analysis on the sparse and imputed sptRNA-seq data.** The total number of significantly enriched clusters (with at least one enriched GO term with FDR adjusted p-value $<$ 0.05) in the 10 tissue sections are shown.

denotes the total number of spots. We then applied K-means on each matrix to partition the genes into 100 clusters. Next, we used the enrichGO function in the R package clusterProfiler [45] to perform the GO enrichment analysis of the gene clusters. The total number of significantly enriched gene clusters (FDR adjusted p-value $<$ 0.05) in each of the 10 tissue sections are shown in Fig 6, which clearly tells that K-means on the imputed sptRNA-seq data produces much more significantly enriched clusters across all the 10 tissue sections than the sparse sptRNA-seq data without imputation.

Finally, we conducted a case study on the Mouse Kidney Section and present the highly relevant functional characteristics in different tissues in mouse kidney detected with the imputations by FIST. For each of the 100 gene clusters generated by K-means as described above, we collapsed the corresponding gene slices in $\tilde{\mathcal{T}}$ into a $n_x \times n_y$ matrix by averaging the slices to visualize the center of the gene cluster. The enrichment results of all the 100 clusters are given in S2 Table. We focus on 3 kinds of representative clusters in Fig 7 which match well with three distinct mouse kidney tissue regions: cortex, ISOM (inner stripe of outer medulla and OSOM (outer stripe of outer medulla). By investigating the enriched GO terms by the clusters (*p*-values shown in Table 3), we found their functional relevance to cortex, ISOM and OSOM regions. We found that the spatial gene cluster 9 which is highly expressed in cortex specifically enriched biological processes for the regulation of blood pressure (GO:0008217, GO:0003073, GO:0008015 and GO:0045777) and transport/homeostasis of inorganic molecules (GO:0055067 and GO:0015672). The spatial gene cluster 23 and 28 which are also highly expressed in cortex enriched cellular pathways that are critical for the polarity of cellular membranes (GO:0086011, GO:0034763, GO:1901017, GO:0032413 and GO:1901380) and the transport of cellular metabolites (GO:1901605, GO:0006520, GO:0006790 and GO:0043648),

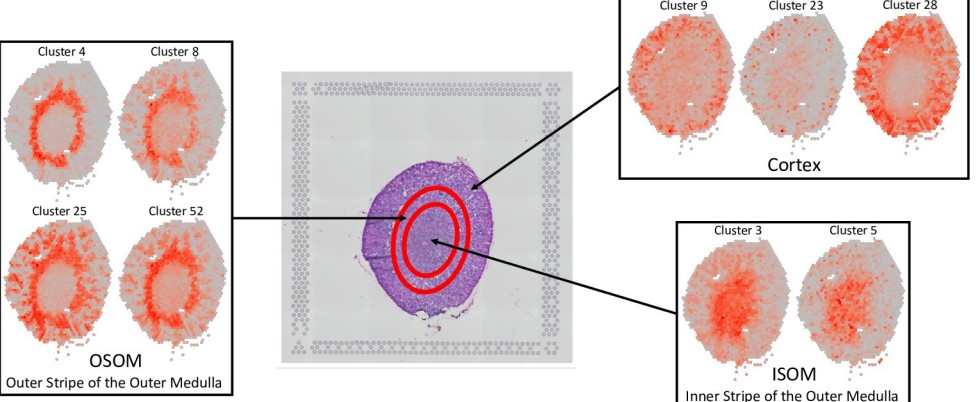

**Fig 7. FIST recovers spatial gene expression patterns on Mouse Kidney Section.** The H&E image of the mouse kidnesy section is shown in the middle with circles roughly separating the tissue area of Cortex, the outer stripe of the outer medulla (OSOM) and the inner stripe of the outer medulla (ISOM) from outer to inner regions. The gene expression patterns of the clusters in each of the three regions are grouped in the same box labeled by the region.

respectively. These observations are consistent with previous studies reporting the regulation of kidney function by the above listed biological processes in cortex [46–49]. In contrast, the pattern analysis of spatial gene expressions in cluster 4, 8, 25 and 52 which are highly expressed in OSOM in kidney showed that catabolic processes of organic and inorganic molecules are specifically enriched such as GO:0015711, GO:0046942, GO:0015849, GO:0015718, GO:0010498, GO:0043161, GO:0044282, GO:0016054, GO:0046395, GO:0006631, GO:0072329, GO:0009062 and GO:0044242. These cellular processes are known to be active in renal proximal tubule which exists across cortex and OSOM [50–55]. Distinctively, the spatial gene clusters highly expressed in ISOM enriched pathways for nucleotide metabolisms (GO:0009150, GO:0009259 and GO:0006163) in cluster 3 and renal filtration (GO:0097205 and GO:0003094) in cluster 5. Collectively, these observations demonstrate that FIST could identify physiologically relevant distinctive spatial gene expression patterns in the mouse kidney dataset. Further, it suggests that FIST can provide a high-resolution anatomical analysis of organ functions in sptRNA-seq data.

## Experiments on additional low-resolution spatial gene expression datasets

To demonstrate that FIST is broadly applicable to impute the spatial gene expression data generated with different platforms, we performed additional experiments on spatial transcriptomics datasets from 3 replicates of mouse tissue (olfactory bulb) provided from an earlier study [56]. Developed before 10x Genomics Visium Spatial protocol, the spatial transcriptomics technology [56] applies an aligned array to profile tissue with both lower spot density and larger spot size (1,007 spots in total, and 200 μm between spots). The design achieves a resolution of 100 μm (10-40 cells per spot). Similar to the experiments on the 10x Genomics data, we organized each of the 3 tissue replicates into a tensor $\mathcal{T} \in \mathbb{R}^{n_p \times n_y \times n_x}$, where $n_x = 33$ and $n_y = 35$ in all the 3 replicates, and $n_p$ is 14,198, 13,818 and 138,40, respectively in replicate 1,2 and 3. The $(i, j, k)$-th entry in $\mathcal{T}$ is the RPKM value of the $i$-th gene at the $(k, j)$-th coordinate in the array.

We performed the spot-wise 5-fold cross-validation as we did in the 10x Genomics data to compare the performances of FIST and the same baseline methods. The distributions of MAE, MAPE and $R^2$ on all the spatial spots in each of the 3 tissue replicates are shown in Fig 8.

**Table 3. Functional terms enriched by spatial gene clusters (most significantly relevant functions).**

| Region | Cluster | Significantly Enriched GO terms |
|---|---|---|
| **Cortex** | **Cluster 9** | GO:0003073—regulation of systemic arterial blood pressure ($p = 9.1 \times 10^{-6}$) |
| | | GO:0008217—regulation of blood pressure ($p = 1.0 \times 10^{-4}$) |
| | | GO:0055067—monovalent inorganic cation homeostasis ($p = 4.3 \times 10^{-4}$) |
| | | GO:0008015—blood circulation ($p = 5.3 \times 10^{-3}$) |
| | | GO:0045777—positive regulation of blood pressure ($p = 5.8 \times 10^{-3}$) |
| | | GO:0015672—monovalent inorganic cation transport ($p = 2.3 \times 10^{-2}$) |
| | **Cluster 23** | GO:0086011—membrane repolarization during action potential ($p = 2.2 \times 10^{-3}$) |
| | | GO:0034763—negative regulation of transmembrane transport ($p = 2.2 \times 10^{-3}$) |
| | | GO:1901017—negative regulation of potassium ion transmembrane transporter activity ($p = 2.4 \times 10^{-3}$) |
| | | GO:0032413—negative regulation of ion transmembrane transporter activity ($p = 2.7 \times 10^{-3}$) |
| | | GO:1901380—negative regulation of potassium ion transmembrane transport ($p = 3.4 \times 10^{-3}$) |
| | **Cluster 28** | GO:1901605—alpha-amino acid metabolic process ($p = 4.8 \times 10^{-10}$) |
| | | GO:0006520—cellular amino acid metabolic process ($p = 6.4 \times 10^{-9}$) |
| | | GO:0006790—sulfur compound metabolic process ($p = 3.1 \times 10^{-6}$) |
| | | GO:0043648—dicarboxylic acid metabolic process ($p = 8.4 \times 10^{-6}$) |
| **OSOM** | **Cluster 4** | GO:0015711—organic anion transport ($p = 7.7 \times 10^{-7}$) |
| | | GO:0046942—carboxylic acid transport ($p = 1.1 \times 10^{-4}$) |
| | | GO:0015849—organic acid transport ($p = 1.1 \times 10^{-4}$) |
| | | GO:0015718—monocarboxylic acid transport ($p = 5.0 \times 10^{-3}$) |
| | **Cluster 8** | GO:0010498—proteasomal protein catabolic process ($p = 1.3 \times 10^{-3}$) |
| | | GO:0006497—protein lipidation ($p = 1.3 \times 10^{-3}$) |
| | | GO:0042158—lipoprotein biosynthetic process ($p = 1.3 \times 10^{-3}$) |
| | | GO:0043161—proteasome-mediated ubiquitin-dependent protein catabolic process ($p = 1.3 \times 10^{-3}$) |
| | **Cluster 25** | GO:0044282—small molecule catabolic process ($p = 5.5 \times 10^{-19}$) |
| | | GO:0016054—organic acid catabolic process ($p = 1.0 \times 10^{-18}$) |
| | | GO:0046395—carboxylic acid catabolic process ($p = 1.0 \times 10^{-18}$) |
| | | GO:0006631—fatty acid metabolic process ($p = 2.9 \times 10^{-16}$) |
| | | GO:0072329—monocarboxylic acid catabolic process ($p = 9.6 \times 10^{-14}$) |
| | | GO:0009062—fatty acid catabolic process ($p = 1.0 \times 10^{-13}$) |
| | | GO:0044242—cellular lipid catabolic process ($p = 4.7 \times 10^{-11}$) |
| | **Cluster 52** | GO:0006732—coenzyme metabolic process ($p = 1.2 \times 10^{-10}$) |
| | | GO:0006520—cellular amino acid metabolic process ($p = 1.6 \times 10^{-10}$) |
| | | GO:1901605—alpha-amino acid metabolic process ($p = 2.3 \times 10^{-9}$) |
| | | GO:0044282—small molecule catabolic process ($p = 2.1 \times 10^{-8}$) |
| | | GO:0000096—sulfur amino acid metabolic process ($p = 2.3 \times 10^{-7}$) |
| **ISOM** | **Cluster 3** | GO:0009150—purine ribonucleotide metabolic process ($p = 7.4 \times 10^{-5}$) |
| | | GO:0009259—ribonucleotide metabolic process ($p = 7.4 \times 10^{-5}$) |
| | | GO:0006163—purine nucleotide metabolic process ($p = 7.4 \times 10^{-5}$) |
| | | GO:0019693—ribose phosphate metabolic process ($p = 7.4 \times 10^{-5}$) |
| | | GO:0072521—purine-containing compound metabolic process ($p = 7.4 \times 10^{-5}$) |
| | **Cluster 5** | GO:0048872—omeostasis of number of cells ($p = 4.5 \times 10^{-5}$) |
| | | GO:0030218—erythrocyte differentiation ($p = 3.2 \times 10^{-3}$) |
| | | GO:0034101—erythrocyte homeostasis ($p = 3.2 \times 10^{-3}$) |
| | | GO:0003094—glomerular filtration ($p = 3.2 \times 10^{-3}$) |
| | | GO:0097205—renal filtration ($p = 3.2 \times 10^{-3}$) |

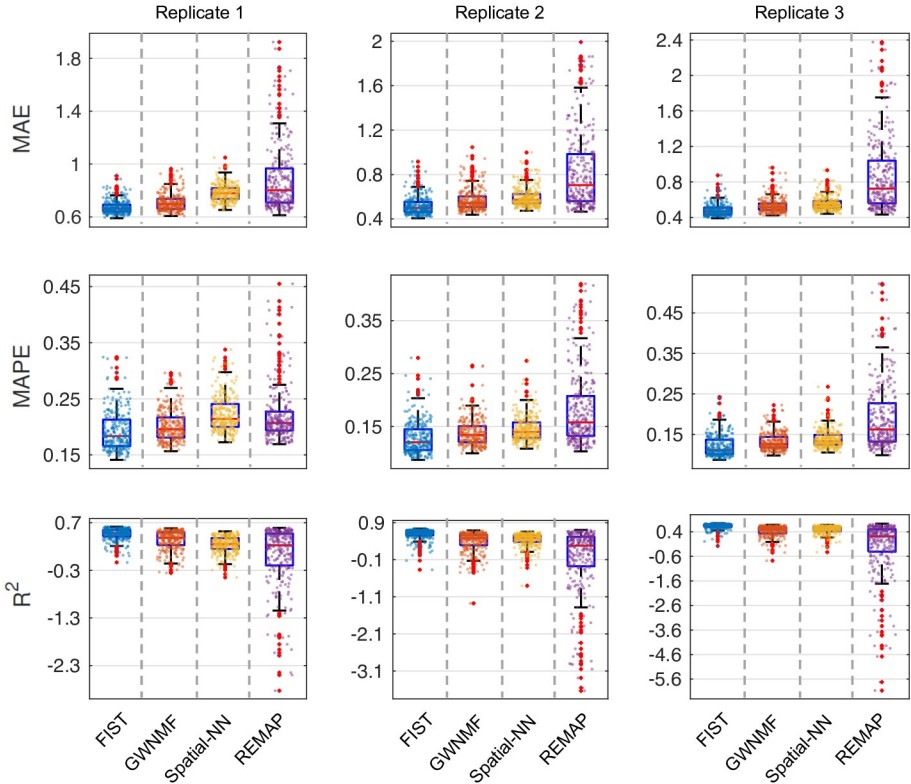

**Fig 8. Spot-wise imputation performance on mouse tissue replicates.** The performances of the four compared methods on the 3 replicates are measured by 5-fold cross-validation. The performance on each spatial spot is denoted by one dot in the box plots. The performances of different methods are shown in different colors.

Consistent with the observations in the previous Figs 2 and 3, FIST clearly outperforms all the baselines with lower MAE and MAPE, and larger $R^2$ in all the 3 replicates. The results suggest that FIST has a potential to be applied to various spatial transcriptomics datasets of different resolution and sparcity to achieves better imputation performance by modeling the spatial data as tensors, and including the prior knowledge with the CPG regularization.

To confirm that the imputation accuracy of FIST is significantly improved by the CPG regularization, we showed in Fig 9 the performance of FIST in each of the 3 replicates by varying the graph hyper-parameter λ in the spot-wise evaluation. It is also consistent with the observation in the previous Fig 4, in which we can observe remarkable reduction on the MAE and MAPE and improvement on $R^2$ by increasing λ to 0.1. The observation also verifies that the CPG topology is informative for the imputation task.

## Discussions

In this study, we propose to apply tensor modeling of multidimensional structure in spatially-resolved gene expression data mapped by the 2D spatial array. To the best of our knowledge, this is the first work to model the imputation of spatially-resolved transcriptomes as a tensor completion problem. Our key observations in the experiments with the ten 10x Genomics Visium spatial transcriptomic datasets are that 1) the imputation accuracy is significantly improved by leveraging the tensor representation of the sptRNA-seq data, and 2) by incorporating the spatial graph and PPI network, the accuracy the imputation and the content of the

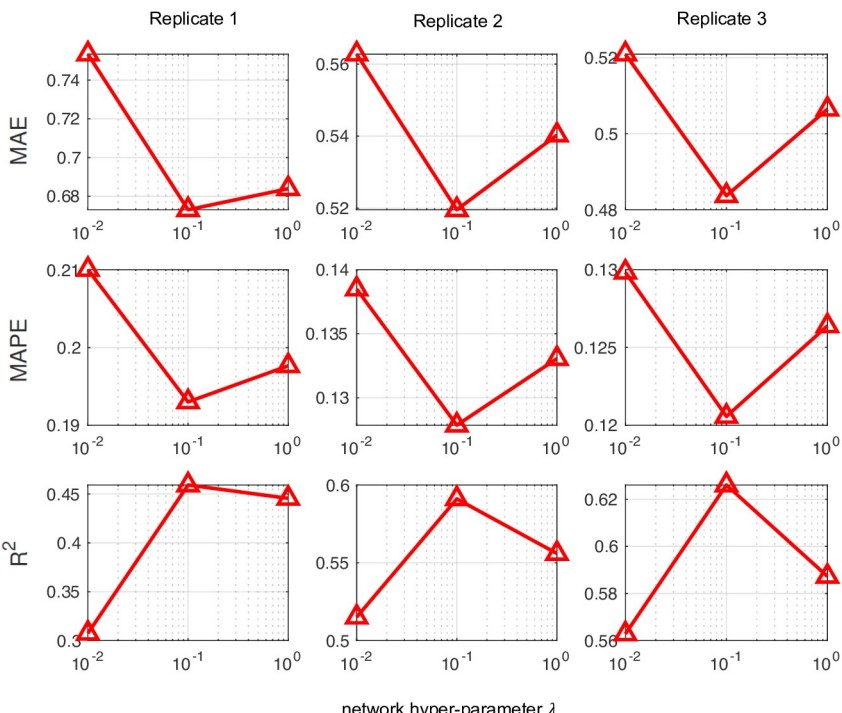

**Fig 9. Imputation performance of FIST on mouse tissue replicates by varying network hyper-parameters.**

functional information in the imputed spatial gene expressions can be further improved significantly.

We observed that the genes that are more sparsely expressed can benefit more from the adjacency information in the spatial graph and the functional information in the PPI network. These genes can be empirically detected with a validation set to tune the only hyper-parameter λ for deciding if the regularization by the product graph is needed for the imputation of a gene. Thus, we expect a low risk of overfitting in applying FIST to other datasets. In addition, the functional analysis of the spatial gene clusters detected on the Mouse Kidney Section data further confirms that FIST detects gene clusters with more spatial characteristics that are consistent with the physiological features of the tissue.

Overall, we concluded that FIST is an effective and easy-to-use approach for reliable imputation of spatially-resolved gene expressions by modeling the spatial relation among the spots in the spatial array and the functional relation among the genes. The imputation results by FIST are both more accurate and functionally interpretable. FIST is also highly generalizable to other spatial transcriotomics datasets with high scalability and only one hyper-parameter needed to tune.

Although our experiments mainly focused on medium density 10x Genomics Visium kit array (5000 spots or 1000 spots), we also plan to further develop variations of FIST for high-resolution spatial transcriptomics datasets with millions of spots generated by high-definition spatial transcriptomics (HDST) [14]. The HDST datasets from the study includes 3 mouse tissue sections from olfactory bulb and 3 human tissue sections from breast cancer using hexagonal array to profile tissue with a high density (1,467,270 spots in total) to achieve a resolution of 2 μm. The imputation tasks on the HDST datasets are quite different for two reasons: First, each cell spans multiple spots. Simple imputation of each spot is not a

well-defined learning problem. Thus, the segmentation of the spots into each individual cell might be necessary as a pre-processing step. Second, the capture efficiency of HDST is as low as 1.3%, which leads to much sparser data for imputation. Our preliminary analysis indicate that the gene expression on the spots are too sparse to be meaningful unless they are aggregated among the spots in a larger region. We plan to develop variations of FIST to overcome these additional challenges.

Another interesting future direction is to develop a variation of FIST for imputing spatial gene expressions with additional information from matched single-cell RNA sequencing data. For example, probabilistic graphical models have been introduced for imputing spatial gene expressions by integration with scRNA-seq data [57]. With the integration of PPI network and tensor modeling, FIST has a great potential to achieve better scalability as well as better accuracy for imputation of transcriptome-wide spatial transcipromics data.

## Supporting information

**S1 Fig. Spatial regions with failed RNA fixing and permeabilization.** The H&E images are shown on the left, and the heatmaps of the total RNA count at each spot are shown on the right. The regions with irregularly low RNA count are annotated by the circles.
(PDF)

**S2 Fig. PPI co-expression analysis.** The Pearson correlation coefficients between expression values of k-hop gene pairs from PPI network are shown as box plots. The Pearson correlation coefficients of different hops are shown in each column.
(PDF)

**S3 Fig. Gene-wise imputation performance by MAE.** The performances on the imputations of each gene are shown as box plots. The MAE of every gene slice is denoted by one dot. The performance of each method is shown in each colored box plot.
(PDF)

**S4 Fig. Gene-wise imputation performance by MAPE.** The performances on the imputations of each gene are shown as box plots. The MAPE of every gene slice is denoted by one dot. The performance of each method is shown in each colored box plot.
(PDF)

**S5 Fig. Gene-wise imputation performance by $R^2$.** The performances on the imputations of each gene are shown as box plots. The $R^2$ of every gene slice is denoted by one dot. The performance of each method is shown in each colored box plot.
(PDF)

**S1 Table. $p$-values of paired-sample $t$-tests.** The means of performance (measured by MAE, MAPE and $R^2$ as in Fig 2) for predicting all the spot fibers are compared between FIST and each of the baseline methods, using paired-sample $t$-tests.
(XLSX)

**S2 Table. Enriched GO terms of spatial gene clusters.** The GO terms significantly enriched by the genes in each spatial gene cluster (FDR adjusted p-value <0.05) are shown in the spreadsheet tables.
(XLSX)

**S1 File. Convergence of FIST.**
(PDF)

## Acknowledgments

We sincerely thank Dr. Kathleen Markham and Dr. Kathleen Greenham for helpful discussion of experiments using 10x Genomics Visium kit.

## Author Contributions

**Conceptualization:** Zhuliu Li, Rui Kuang.

**Data curation:** Zhuliu Li, Tianci Song, Rui Kuang.

**Formal analysis:** Zhuliu Li, Tianci Song, Rui Kuang.

**Funding acquisition:** Rui Kuang.

**Investigation:** Zhuliu Li, Tianci Song, Jeongsik Yong, Rui Kuang.

**Methodology:** Zhuliu Li, Rui Kuang.

**Project administration:** Rui Kuang.

**Resources:** Rui Kuang.

**Software:** Zhuliu Li.

**Supervision:** Rui Kuang.

**Validation:** Zhuliu Li, Tianci Song, Jeongsik Yong, Rui Kuang.

**Visualization:** Zhuliu Li, Tianci Song, Jeongsik Yong.

**Writing – original draft:** Zhuliu Li, Tianci Song, Jeongsik Yong, Rui Kuang.

**Writing – review & editing:** Zhuliu Li, Tianci Song, Jeongsik Yong, Rui Kuang.

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
