## [Decision Letter · Decision Letter 0]

6 Nov 2020

Dear Prof. Kuang,

Thank you very much for submitting your manuscript "Imputation of Spatially-resolved Transcriptomes by Graph-regularized Tensor Completion" for consideration at PLOS Computational Biology.

As with all papers reviewed by the journal, your manuscript was reviewed by members of the editorial board and by several independent reviewers. In light of the reviews (below this email), we would like to invite the resubmission of a significantly-revised version that takes into account the reviewers' comments.

We cannot make any decision about publication until we have seen the revised manuscript and your response to the reviewers' comments. Your revised manuscript is also likely to be sent to reviewers for further evaluation.

Sincerely,

Sushmita Roy, Ph.D.

Associate Editor

PLOS Computational Biology

Jason Papin

Editor-in-Chief

PLOS Computational Biology

Reviewer's Responses to Questions

**Comments to the Authors:**

Reviewer #1: Reproducibility report has been uploaded as an attachment.

Reviewer #2: REVIEW IS UPLOADED AS AN ATTACHMENT.

Reviewer #3: The paper presents an approach to imputation of gene expression in spatial transcriptomics data using graph-regularised tensor completion. This is an interesting class of model for spatial data and I can imagine that it could be useful in smoothing data or in other data analysis tasks, e.g. smoothing followed by clustering may well be a good approach. I can also believe that the method performs well in the synthetic benchmarks where data are removed (at random I think?) and then imputed, compared to the other models considered here. However, it was not clear to me exactly how it should be used in real data. Are the authors suggesting that all zeros should be imputed with non-zero values? Is this what is done (I was not sure) and is that justified? The zeros are not missing-at-random data, they are data that happens to be zero usually because expression levels are low and read-depth is limited. Replacing them with non-zero values seems to me to be more akin to smoothing than to imputation. It is therefore not clear to me that this is justified. Smoothing may be useful prior to modelling (e.g. clustering) but that is not the same as imputation which should really only be used to deal with missing data, not low counts data. I'm afraid I therefore don't agree with the way the method is presented as an imputation approach. I have two specific major concerns:

Major comments:

[1] It is not clear in modern UMI-normalised data whether (or when) zeros should be considered artefacts to be corrected/imputed since they may simply reflect genuinely weak expression. If the expression is weak then imputation should not be used and will smooth out genuine signal in the data. If genes are weakly expressed then counts can be very low and zeros simply reflect read depth limitations. For example, in the case of single-cell RNA-Seq data recent work shows that UMI-normalised data is not highly zero-inflated [1,2,3] and this was really only a problem on older datasets without UMI normalisation where amplification artefacts were harder to remove. I imagine the same holds true for UMI-normalised spatial transcriptomics data, in which case imputation of zero-valued genes is not justified. The citations in the introduction discussing modelling zero-inflation (Refs [22, 25, 26. 27] in the paper) predate these more recent works and often model zero-inflation as an artefact to be corrected for rather than simply part of a standard measurement model such as a negative binomial distribution over counts.

[2] Another major concern I have it that PPI data is used to regularise genes on the basis that interacting genes are more likely to be correlated in gene expression. I was wondering how good is this assumption since it seems very easy to check with data. Another approach would be to use previous expression datasets to assess co-expression, for example, since expression data are so plentiful. Is PPI the best choice for determining whether genes are likely correlated? This could be tested on observed expression data, e.g. how much co-expression is explained by PPI data?

References

[1] Choi, K., Chen, Y., Skelly, D. A., & Churchill, G. A. (2020). Bayesian model selection reveals biological origins of zero inflation in single-cell transcriptomics. bioRxiv.

[2] Svensson, V. (2020). Droplet scRNA-seq is not zero-inflated. Nature Biotechnology, 38(2), 147-150.

[3] Sarkar, A. K., & Stephens, M. (2020). Separating measurement and expression models clarifies confusion in single cell RNA-seq analysis. BioRxiv.

**Have all data underlying the figures and results presented in the manuscript been provided?**

Reviewer #1: None

Reviewer #2: Yes

Reviewer #3: Yes

PLOS authors have the option to publish the peer review history of their article (what does this mean?). If published, this will include your full peer review and any attached files.

Reviewer #1: No

Reviewer #2: No

Reviewer #3: No
---

## [Decision Letter · Decision Letter 1]

14 Mar 2021

Dear Prof. Kuang,

Thank you very much for submitting your manuscript "Imputation of Spatially-resolved Transcriptomes by Graph-regularized Tensor Completion" for consideration at PLOS Computational Biology. As with all papers reviewed by the journal, your manuscript was reviewed by members of the editorial board and by several independent reviewers. The reviewers appreciated the attention to an important topic. Based on the reviews, we are likely to accept this manuscript for publication, providing that you modify the manuscript according to the review recommendations. 

In particular please address reviewer 2's comment.

Sincerely,

Sushmita Roy, Ph.D.

Deputy Editor

PLOS Computational Biology

Jason Papin

Editor-in-Chief

PLOS Computational Biology

[LINK]

Reviewer's Responses to Questions

**Comments to the Authors:**

Reviewer #1: Reproducibility report has been uploaded as an attachment.

Reviewer #2: The authors have satisfied my concerns, they have:

1) satisfied my request of broad applicability by performing "additional experiments on the spatial transcriptomics datasets from 3 replicates of mouse tissue (olfactory bulb) provided by Stahl et al. (2016)."

2) clarified and articulated the imputation problems. They performed both 1) spot-wise imputation (newly added) and 2) gene-wise imputation experiments.

3) added additional information to run the scripts. "We provided the scripts to display the key results reported in the paper."

4) clearly highlighted the scalability of the method. "we also included the space complexity of FIST together with the time complexity to justify that FIST is a scalable methods without need of computation with the full CPG."

One more minor point that does not need to return to me for acceptance:

- In the abstract it reads: "FIST significantly outperformed several best performing single-cell RNAseq data imputation methods."

"several best performing methods" sounds like an oxymoron. There is only one best performing method, not several.

Reviewer #3: The authors have addressed all my comments in the revised version

**Have all data underlying the figures and results presented in the manuscript been provided?**

Reviewer #1: Yes

Reviewer #2: Yes

Reviewer #3: Yes

PLOS authors have the option to publish the peer review history of their article (what does this mean?). If published, this will include your full peer review and any attached files.

Reviewer #1: **Yes: **Anand K. Rampadarath

Reviewer #2: No

Reviewer #3: No

Figure Files:

Data Requirements:

Reproducibility:

References:

---

## [Editor Report · Decision Letter 2]

19 Mar 2021

Dear Prof. Kuang,

We are pleased to inform you that your manuscript 'Imputation of Spatially-resolved Transcriptomes by Graph-regularized Tensor Completion' has been provisionally accepted for publication in PLOS Computational Biology.

Best regards,

Sushmita Roy, Ph.D.

Deputy Editor

PLOS Computational Biology

Jason Papin

Editor-in-Chief

PLOS Computational Biology

---

## [Editor Report · Acceptance letter]

30 Mar 2021

PCOMPBIOL-D-20-01355R2 

Imputation of Spatially-resolved Transcriptomes by Graph-regularized Tensor Completion

Dear Dr Kuang,

I am pleased to inform you that your manuscript has been formally accepted for publication in PLOS Computational Biology. Your manuscript is now with our production department and you will be notified of the publication date in due course.

With kind regards,

Andrea Szabo
